# What Drives Land Abandonment in Core Grain-Producing Areas? Evidence from China

**DOI:** 10.3390/ijerph19095090

**Published:** 2022-04-22

**Authors:** Yumeng Wang, Jiaxu Li, Xiangzhi Kong

**Affiliations:** School of Agricultural Economics and Rural Development, Renmin University of China, Beijing 100872, China; wymmyw@ruc.edu.cn (Y.W.); lijx995773@163.com (J.L.)

**Keywords:** explicit arable land abandonment (EALA), implicit arable land abandonment (IALA), pathway analysis, fuzzy set qualitative comparative analysis (fsQCA)

## Abstract

Food security remains a major issue for developing countries. Reducing arable land abandonment (ALA) is crucial to ensuring food security. In China, the ‘decline in both quantity and quality’ of arable land resources, especially in major grain-producing areas, has become increasingly serious. This study uses fuzzy set qualitative comparative analysis (fsQCA) to explore the core conditions and combinations of paths leading to explicit and implicit abandonment using 30 typical cases in the main grain-producing areas of Hubei Province. The results show that (1) three combined pathways lead to explicit ALA (EALA) and that two pathways lead to implicit ALA (IALA); (2) laborer health (LH) is the core condition leading to EALA; and (3) LH, agricultural laborer (AL), per capita income (PCI) and social relationships (SRs) are the core conditions leading to IALA. To effectively alleviate ALA, the government should improve production conditions, pay attention to laborer health issues, improve agricultural returns and strengthen food security publicity and guidance, thereby promoting the rational use of arable land in these areas. The findings in this study link the changes in arable land use and provide a reference for other developing countries in ensuring food security.

## 1. Introduction

Arable land abandonment (ALA) is one of the dominant processes of change in rural areas. Globally, ALA emerged in the early 20th century and has increased since the 1950s. In some cases, ALA has led to some positive effects. Examples of these positive effects include increased carbon sequestration [1], improved water quality [2], the restoration of ecosystem self-maintenance, and promotion of biodiversity [3]. However, it has had significant negative impacts, including species invasion [4,5], an increased risk of wildfires [6], the degradation of agricultural landscapes, and the decay of villages, which in turn have resulted in the loss of traditional farming culture and aesthetic values [7] and a decline in tourism attractiveness. More notably, abandoned agricultural land reduces the number of crop sowing areas, which may lead to a substantial reduction in grain output in some areas [8] and may lead to problems such as grain shortages in abandoned areas [9,10]. Although the global grain output has increased in recent decades, in the post-COVID-19 era, the domestic agricultural labor force has shrunk, grain output has been reduced, and the instability and imbalance of the international grain trade market has intensified [11]. Some countries have formulated policies such as banning grain exports to ensure their own food security. Methods by which to solve the national food security problem stably and sustainably are still key to the survival and development of China. Furthermore, it is also worth paying attention to the reduction in food production that has been caused by ALA.

ALA in China includes both explicit ALA (EALA) and implicit ALA (IALA). In the main grain-producing regions, where strict agricultural land protection policies are in place, it is rare for ALs to leave their land unused. Instead, there is a widespread tendency to leave agricultural land unoccupied. The factors that lead to ALA are complex, but if IALA were to occur on a large scale in major food-producing areas, then it would not only pose a threat to national food security but would also affect China′s socioeconomic development. Therefore, agricultural and land management policy makers must distinguish between the factors that lead to EALA and those that lead to IALA by ALs.

Although some scholars have studied the factors that lead to ALA, for instance, urbanization [12], demographic changes [13], declining farm profit and the increasing cost of farm inputs [14,15], few studies have examined the effects of different combinations of factors on the EALA and IALA of agricultural households [16]. Therefore, we propose the use of fsQCA to elucidate the EALA and IALA situation in Louhe township, Hubei Province. Specifically, we identify the ideal core conditions and combination pathways, explain the key condition variables that lead to cultivated land abandonment, and discuss the different combinations of conditions that exist and are causative of EALA and IALA. The aim is to provide a reference for mitigating the phenomenon of ALA in China and to improve the efficiency of arable land resource use.

This study contributes to the literature in three ways. First, it provides insight into ALA activities by exploring the effect of the interdependence of multiple factors on EALA and IALA among ALs. Moreover, this work aims at increasing the understanding of the complex relationships between various institutions and ALA activities. Second, this study proposes an integrative framework for analysing ALA activities across ALs. Unlike previous studies, which focused mainly on the net effects model of causality between ALA and conditional variables, such as arable land distance (ALD), arable land terrain (ALT), irrigation conditions (ICs), agricultural laborers (ALs), laborer health (LH), per capita income (PCI), and social relationships (SRs), this study incorporates EALA and IALA institutions into an analytical framework based on a configurational perspective. This framework helps to identify the causal complexities between various conditions and ALA activities. Third, this study contributes to institutional theory by expanding its application in situations of causal complexity. Some scholars have categorized its various factors into three dimensions, such as population, social economy, and production conditions, thereby providing a theoretical foundation for understanding ALA activities. Although there are connections among these dimensions, the interdependence of institutions on the focal phenomenon has rarely been considered in the literature, and is constrained by conventional statistical methods. Thus, this study employs the fsQCA configuration approach to expand the application of theory to analyse the complex relationships between multiple factors and ALA activities.

The remainder of the paper is structured as follows. The next section provides a brief literature review. Then, seven conditional variables are initially identified, an empirical study design based on fuzzy set qualitative comparative analysis (fsQCA) is proposed, the main combinatorial configurations of the conditional variables and their underlying mechanisms are examined, the main findings are explored, and policy recommendations for the effective management of both EALA and IALA in major food-producing regions are proposed.

## 2. Theoretical Analysis

### 2.1. Definition of Arable Land Abandonment

ALA, also known as ‘agricultural land abandonment’ and ‘cropland abandonment’, refers to the cessation of farming and allocation of land for natural successions, such as grasses, shrubs, and trees on former agricultural lands, but may also result in land degradation [8]. ALA can be divided into two forms: EALA and IALA. The former refers to the state of land when it is left idle or unmanaged [17,18], while the latter refers to the state in which arable land and the labor force are combined but insufficient [19,20]; for example, when investment in fertilization or labor is reduced, and the planting pattern of two crops per year changes to one crop per year. Rudel suggests that changing the use of arable land in exchange for other forms of agricultural economic activities can also be regarded as IALA [21]. Combined with previous research and the actual investigation situation of the study area, this paper defines ALA as the phenomenon of idle or underutilized land caused by the abandonment of cultivation for various reasons and includes not only the EALA of completely idle arable land but also the IALA of underutilized cultivated land due to the reduced investment in labor and capital.

### 2.2. Drivers of Arable Land Abandonment

The occurrence of ALA is influenced by many factors. Developed regions such as Europe, North America, Australia, and Japan have the most widely distributed areas of ALA [22]. ALA in Europe and North America is related mainly to the decline in market demand [23], institutional change [24], land fertility [25], and climate conditions [22]. ALA in Japan is related mainly to population ageing and population loss [26]. In developing countries undergoing rapid industrialization and urbanization, such as the Philippines, Nepal, and Thailand, ALA is related to the rising opportunity cost of agricultural labor, agricultural labor distance and insufficient labor force [27].

### 2.3. Explanation of the Factors Affecting Arable Land Abandonment

ALA is a kind of behaviour-related decision making, with ALs weighing the benefits of agricultural labor and working based on their arable land production conditions in the context of rural population outflow caused by urbanization and industrialization. Moreover, ALA is affected by many factors, such as population, social economy, and production conditions. The formation process of abandonment is synchronized with the outflow of the agricultural labor force in the context of regional economic development and industrial structure adjustment. Combined with the relevant literature, we have classified the factors influencing ALA into seven categories: ALD, ALT, IC, AL, LH, PCI and SRs. We explored the multiple concurrent causes, as well as their complex mechanisms, that influence ALA through these seven antecedent conditions. In the following section, we elaborate on the linkages between these seven conditions and ALA in more detail.

ALD is the distance between arable land and residential houses. The farther the arable land is from habitations, the greater the possibility of abandonment because the increase in this distance significantly increases the time cost of arable land operation and makes it more likely to be abandoned. Based on the plot scale, Shi and Li studied the relationship between ALD and the abandonment rate of arable land and found that the greater the ALD, the higher the abandonment rate of arable land and that 500 m and 900 m are the two turning points [28]. The abandonment rate of arable land with a cultivation distance of <500 m fluctuated mainly in the range of 15–20%, while that of arable land with a cultivation distance of 500–900 m fluctuated mainly in the range of 20–35%, and that of arable land with a cultivation distance of >900 m increased sharply to more than 40%. Moreover, Li et al. found that the transportation of fertilizer and grain in the study area relied overwhelmingly on the agricultural labor force, most of whom are over 60 years old, with elderly individuals and women being the main actors [29]. With the observable increase in ALD, the transportation time spent by ALs increases, transportation difficulty intensifies, and the proportion of abandonment increases. In the case of labor scarcity, arable land that is distant from residential houses, near forested land, or in hard-to-access places is relatively unlikely to be cultivated [17,30].

ALT is characterized by a fluctuation in the surface shape of arable land. The more uneven the terrain is, the greater the possibility of abandonment because terrain conditions can limit agricultural machinery inputs [31]. With the precipitation of the agricultural labor force and ageing population, poor-quality arable land with rugged terrain continues to be marginalized and eventually abandoned [8,32]. In addition, global studies have shown that steeper arable land is at a higher risk of erosion and landslides [33]. Therefore, arable land with a rugged and undulating topography and ecological fragility is more likely to be abandoned before arable land characterized by flat terrain [34].

ICs are technical facilities that provide the water required for proper crop growth. To some extent, ICs can determine the quality of arable land and directly affect the abundance and failure of agricultural production [35]. In economically underdeveloped areas, irrigation on abandoned arable land is generally not high, and more than 70% of abandoned arable land is distributed in areas that lack irrigation facilities. In addition, the above study found that well-irrigated arable land was less likely to be abandoned than plots that could not be irrigated [14]. The probability of abandonment was reduced by approximately 10% when arable land had access routes to irrigation water, which suggests that investment in irrigation infrastructure and improved access to irrigation for ALs can help to reduce the extent of land abandonment [36].

ALs are engaged in agricultural production activities. In recent years, with the rapid development of industrialization and urbanization, the gap between the high wages of nonagricultural employment and the low income of small-scale agricultural labor operations has become increasingly prominent, inducing a large number of ALs to shift to nonagricultural employment [37]. From the perspective of labor supply and demand, the unbalanced allocation of two factors of production, arable land and labor, can lead to the occurrence of land abandonment [38]. For example, among the ALs who abandoned land in the main grain-producing area of the Poyang Lake Plain, Li et al. found that 30.93% of them indicated that the reason for their abandonment was a labor shortage [39]. Ayambire et al. argued that the advantages of agricultural labor are greatly reduced due to insufficient labor, a left-behind labor force, and relatively high levels of per capita arable land [40]. Therefore, the smaller the agricultural labor force is, the greater the possibility of ALA.

LH refers to the status of an individual’s morphological structure, physical quality, and mental outlook. Health is a basic prerequisite for people to engage in production and life, and it plays an important role in land use [41]. Although good ICs and mechanization have made food cultivation much less labor intensive, agricultural work that requires healthy, young and strong laborers to perform heavy physical and seasonal work can be performed by laborers of weaker health. However, the special nature of agricultural labor and the hardship of the labor environment determine that a healthy body is the first factor of production, thus if the quality of labor supply is decreased, households may decrease or abandon their labor input in arable land management [42]. In other words, the higher the level of health status is, the less ALA there is.

PCI is the average income of agricultural households according to their household size, which can reflect their average income level. In the survey sample, Vellema et al. found that lots of agricultural households derive their main income from industries other than agriculture [43]. The significant and positive coefficient of household PCI on ALA implies that the more labor that is allocated to nonagricultural activities, thus reducing the household′s agricultural labor, the higher the possibility of ALA. However, Martin and Clapp’s theoretical study suggested that higher income increases ALs′ investment in agriculture, with activities such as adopting advanced agricultural technologies and purchasing more agricultural machinery, which can effectively alleviate the abandonment phenomenon [44]. Thus, the effect of PCI on ALA is uncertain and may either accelerate or reduce abandonment.

SRs are the social resources that ALs have to sustain their livelihoods, pursue their own development and cope with risk shocks. Moreover, SRs refer essentially to the ability of an individual or organization to build SRs to access resources through trust, participation, and prestige. An empirical study conducted by Deng et al. revealed that if an AL has a relative who is a village official, then it implies that he or she has some influence in the village [45]. These ALs have better access to technology, information and financial help and are more likely to engage in large-scale agricultural production. Ayele and Degefa, however, found that the social resources of ALs increased their willingness to migrate to other locations for employment [46]. In rural China, Xiong, Zhu, and Zhang showed that abundant social resources play an important role in helping Als to enter the nonagricultural sector, thereby increasing the probability of their ALA [47]. However, the mechanisms and effects behind the differences in SRs on ALA behaviour are unclear.

By combing the existing studies, we investigated how these seven conditional variables affect ALs′ ALA behaviour. However, we did so mainly from a single perspective, taking a single factor or a single behaviour as the study object, with the aim to explore the role of the relationship between that factor and ALA. Few studies have addressed the effects of a combination of multiple factors and the way in which multiple factors briefly complement and reinforce each other or coordinate and match each other in terms of ALA. Therefore, it is necessary to combine the influencing factors to analyse the process of ALA in greater depth. In addition, there are individual conditions that are uncertain in terms of their effect on land abandonment, so their driving mechanisms in a qualitative comparative analysis depend on a combination of other conditions.

## 3. Methods

### 3.1. Study Areas

Louhe town in Xiantao city was selected as the study area (illustrated in Figure 1 and Figure 2), which is in the main grain-producing area and experienced a large population migration in the past ten years. Louhe town is in Jianghan Plain, which has abundant water and heat resources and is suitable for large-scale crop planting. The climate and soil conditions facilitate the planting pattern of two crops per year. Different forms of ALA have varying impacts on grain production. This town has a subtropical continental monsoon climate, with four distinct seasons, including hot and rainy seasons. The annual average temperature is 17.0 °C, the annual average precipitation is 1238.6 mm, and the annual average hours of sunshine is 1877.8 h. The area is flat with fertile soil, sufficient light, heat, and water resources, and has excellent agricultural production conditions. The main crops are rice, rape, cotton, soybeans, etc. Considering the regional differences in agricultural production and ALA in 15 towns of Xiantao city, Louhe town can represent and reflect the overall ALA situation and characteristics.

Louhe town in Xiantao city covers an area of 17,530 km, arable land accounts for 48%, 38 administrative villages, and the total population was 82,769 at the end of 2020. Agricultural resources and water resources are abundant, and crops mainly include rice, wheat and rape. Seven typical villages were selected, including Hetai, Zhongzhou, Huangqiao, Dengling, Shuangqiao, Wangchang and Xinchang.

### 3.2. Data Sources

Firsthand data were collected using a household survey conducted at the end of August 2021 in Louhe town, Xiantao city, Hubei Province. The pre-study questionnaire was verified, analyzed, and then modified and improved because of the existing questionnaire. In September 2021, a questionnaire survey for participatory assessment was conducted in the homes of ALs in the study area. ALs were selected from each village according to the actual ALA situation, covering different types of agricultural households. Although we did not follow strict random sampling when selecting cases, we still strove to cover all ALA types with regard to the specific land use situations, plots’ characteristics, rural households’ characteristics and so on. Considering the important role performed by and typical representation of the heads of households, we strictly controlled survey respondents so that only heads of households answered the questions. Each household was surveyed at random, and a total of 34 questionnaires were obtained. Some samples were excluded according to the completeness of the data, and finally, 30 valid questionnaires were obtained. The main contents of the questionnaires included the basic characteristics of agricultural households (including age, health, income, education level, nonagricultural status, and social resources), the basic agricultural labor resource status of agricultural households (area of arable land, type of arable land, cultivation structure, ICs, geographical location, etc.), the status of the arable land resource utilization of agricultural households (area of abandonment, type of abandonment, possible influencing factors, etc.), the status of agricultural production and operation (labor input, degree of mechanization, fertilizer and pesticide input, etc.), and mechanization (chemical fertilizer, pesticide inputs, land output, etc.).

### 3.3. Fuzzy Set Qualitative Comparative Analysis

fsQCA aims to establish connections between combinations of causal conditions. In this study, seven factors were selected, namely ALD, ALT, ICs, ALs, LH, PCI, and SRs, and an outcome (i.e., ALA). The results of the analysis show which combinations of conditions are related to an outcome of interest [48].

The fsQCA process includes several phases [49]. In the first phase, the raw data are calibrated and transformed into fuzzy set data, with the process reported transparently. In the second phase, a ‘truth table’, which is a logic-based mathematical table and reports all the possible combinations of casual conditions associated with an outcome, is generated. In the third phase, the number of rows in the truth table is reduced considering the frequency and consistency threshold. For this study, we used a ‘frequency threshold’ of 1 and a ‘consistency threshold’ of 0.80 [50]. In the fourth phase, an algorithm is used to simplify the truth table, and the solution is minimized and analyzed. In this phase, the authors must define how to logically handle the reminders by using one of the three following techniques: a parsimonious solution, intermediate solution or complex solution. The parsimonious solution includes all simplifying assumptions, the intermediate solution includes simplifying assumptions by including easy counterfactuals, and the complex solution does not include easy and difficult counterfactuals [49].

### 3.4. Measures and Calibration

An important step in preparing the dataset for fsQCA is calibration, which is a process through which cases are assigned set membership scores. We adopted both three- and four-value schemes to calibrate the measurements of the outcomes and causal conditions. When external standards could be implemented using specified values from an interval scale, we used the three-value scheme corresponding to the following three key breakpoints: (a) fully in membership (the upper bound of the set membership of a case in a fuzzy set, assigned a value of 1), (b) fully out membership (the lower bound of the set membership of a case in a fuzzy set, assigned a value of 0), and (c) a crossover point (that is, the point of maximum ambiguity and that is neither in nor out of a particular set, assigned a value of 0.5) [49,51]. If the information was not systematic or strictly comparable between cases [49], then we used the four-value scheme, which marks ‘more in than out’ at 0.67 and ‘more out than in’ at 0.33, in addition to fully in membership (set as 1) and fully out membership (set as 0). The calibration anchors are reported in Table 1.

For the assignment of the outcome variables, we assessed each respondent’s membership using a four-value fuzzy set, for example, EALA (1) as fully in membership, IALA (0.67) as more in than out in the set membership of frequency, the respondent not abandoning land but having a willingness to abandon in the future (0.33) as more out than in the set membership of frequency, and the respondent not abandoning land and having no tendency to abandon in the future (0) as fully out the set membership.

ALD is an important factor affecting the ALA, and in general, the farther the arable land is from the AL′s residence, the higher the probability of the arable land being abandoned [28,31]. We use the number of metres (m) between arable land and residential houses as the conditional variable. With reference to Fiss [51] and Greckhamer et al. [52], we adopted an adjusted distribution method to calibrate ALD. Following Greckhamer et al. [52] with a stricter standard on data distribution (i.e., the 5th, 50th, and 95th percentiles), we initially selected the crossover point at 367.5 (approximately the 50th percentile) and decided upon 914 as a high full-membership anchor (approximately the 95th percentile). Accordingly, the full nonmembership anchor was set at 59.45 (approximately the 5th percentile).

Based on prior studies on ALT, in areas with higher elevations, steeper slopes, poor soil conditions, and poor field facilities, cultivated land abandonment is more likely to occur [4,25,53]. We calibrated ALT by grouping the terrain into the following four categories: low lying as fully in membership (1), relatively low lying as more in than out (0.67), relatively flat as more out than in (0.33), and flat as fully out membership (0).

In this study, ICs were assessed according to the degree of the abundance of irrigation facilities. Insufficient irrigation facilities were set as fully in membership (1.00), relatively insufficient irrigation facilities were set as more in than out at (0.67), relatively sufficient irrigation facilities were classified as more out than in (0.33), and those that have abundant irrigation facilities were classified as fully out membership (0).

There are many migrant workers in the research area, with 1 or 2 ALs working on each parcel of agricultural land, and there are also some part-time laborers. Referring to Li et al. [29], the calculation method for the number of people within the labor force engaged in agricultural and nonagricultural activities (for pure agricultural labor only), is as follows: agricultural labor force is counted as 1 for agricultural labor with individuals only engaged in agricultural labor, the agricultural labor force is counted as 0.2 for a labor force working in other areas while also being engaged in agricultural labor, agricultural labor force is counted as 0.5, and the nonagricultural labor force is counted as 0. We initially selected the crossover point at 1.2 (approximately the 50th percentile) and decided upon 2 as a high full-membership anchor (approximately the 95th percentile). Accordingly, the full nonmembership anchor was set at 0 (approximately the 5th percentile).

We measured labor health by observing the health status of respondents and their responses. In this set, respondents with more serious diseases, such as tumours and impaired body functions, are coded as being fully in (1.00); those who are overworked and not feeling well, which affects their agricultural activities, are coded as being more in than out in the set membership of transferability (0.67); those who have some common underlying diseases, such as hypertension, but can be stabilized by taking medication are coded as being more out than in the set membership of transferability (0.33); and those not suffering from diseases are coded as fully out (0).

PCI can reflect the average income level of agricultural households. We adopted the adjusted distribution method to calibrate PCI, chose the crossover point as 2732.2 (approximately the 50th percentile) and decided upon 5776 as a high full-membership anchor (approximately the 95th percentile). Accordingly, the full nonmembership anchor was set at 878.5(approximately the 5th percentile).

Essentially, SRs identifies the ability of individuals to increase their access to resources, which includes whether respondents have relatives who are village officials. The answer options are ‘Yes’ or ‘No’ and were then translated into a binary variable, where yes = 1 and no = 0.

We used fsQCA 3.0 for the calibration and subsequent analysis. A constant of 0.001 was added to variables under 1 to avoid theoretical challenges related to the 0.5 membership score [51]. Table 2 presents the results after calibration.

## 4. Results

### 4.1. Necessity Analyses

Before performing the group analysis, the necessity of a single condition needed to be tested. In general, when the consistency of a condition variable is ≥0.9, the variable can be considered necessary for the occurrence of the result. Table 3 reports the results of the necessity test for individual condition variables. As seen from Table 3, the consistency of each individual antecedent condition variable did not reach the threshold level of 0.9, which implies that the explanatory power of ALD, ALT, ICs, ALs, LH, PCI and SRs in terms of ALA is weak. In other words, the abandonment of arable land is not due to individual conditions. This finding also indirectly suggests that ALA is the result of the synergistic effect of multiple factors. Therefore, it is necessary to further present a combination of the linkage matching effects among variables in order to obtain more information on the pathways involved in the occurrence of ALA.

### 4.2. Sufficiency Analyses

Next, the antecedent condition histories of ALA were further analysed. According to prior studies [51], we established a raw consistency threshold at 0.8 to conduct a sufficiency analysis [54,55]. We set the proportional reduction in inconsistency (PRI) score to 0.7 to reduce contradictory configurations [52]. As the sample is medium sized and follows prior studies [49], we set the frequency threshold at one case per configuration.

fsQCA produces three kinds of solutions, which are as follows: complex, intermediate, and parsimonious. In this study, the intermediate solution, complemented by the parsimonious solution, was mainly reported and we then interpreted the results [51]. The intermediate solution incorporates easy counterfactuals (i.e., those consistent with theoretical or substantive knowledge) [52]. The parsimonious solution considers difficult counterfactuals (i.e., those inconsistent with theoretical or substantive knowledge) [52]. Together, these two solutions can distinguish between core and peripheral conditions [55]. Core conditions are consistently present in intermediate and parsimonious solutions; peripheral conditions appear only in the intermediate solution. Core conditions are key components because they remain in the solution even with difficult counterfactuals, whereas peripheral conditions are stripped away in such cases [52]. Table 4 shows the results of the configuration analysis of the seven conditions formed by the pathway(s) for EALA and IALA.

The five pathways presented in Table 4 present a level of consistency that is higher than the minimum acceptable standard of 0.75 for both individual solutions (pathways) and the overall solution, and the overall solution had a consistency of 0.925. The overall solution was found to have a coverage of 0.387, and the five pathways weighted in Table 4 can be considered as a sufficient combination of EALA and IALA by ALs.

### 4.3. Supplemental Analyses

By looking at each pathway individually (longitudinal), in pathway 1, the presence of ALD, ALT, ICs and LH, and the absence of PCI and SRs can be seen to play a central role, with the presence of ALs playing a supporting role, the consistency of which is the highest (1). In pathway 2, the presence of ALT, ICs and LH and the absence of an AL contribute to the core condition, while the presence of PCI and SRs and the absence of ALD constitute the auxiliary condition, and the unique coverage of this group is 0.041, covering one case. In pathway 3, the presence of ALD and LH, the absence of an AL, SRs, and PCI are core conditions, and the absence of ALT and ICs are auxiliary conditions, covering two cases.

In pathway 4, the presence of an AL, PCI and SRs and the absence of LH are the core conditions, and the absence of ALD, ALT, and ICs are auxiliary conditions. The lowest consistency (0.902) is found for pathway 4, but its unique coverage is the highest (0.122), covering three cases. Pathways 5 and 4 have the same core conditions but differ in terms of auxiliary conditions, with the absence of ALD and the presence of ALT and ICs playing an auxiliary role. To address these complex relationships, an in-depth explanation of the five pathways listed above is provided by combining theory and in-depth case studies in the ‘Theoretical explanation and case study’ section.

### 4.4. Sensitivity Analyses

We conducted a sensitivity analysis to evaluate the robustness of the results, adopting the method of adjusting the consistency threshold level as proposed by Ragin [56]. Without changing the frequency of cases, the consistency level of the robustness test was improved by 0.05 [57]. In the above study, the concordance was adjusted from 0.7 to 0.75 and it was found that the adjustment of the parameters did not lead to substantial changes in the number of configurations, components, and fit parameters of concordance and coverage. Therefore, the analysis results were considered relatively reliable.

### 4.5. EALA Pathways Analyses

Pathway 1 shows that when ALs are in poor health and the arable land is far away from their homes, low lying and lacking in irrigation facilities, combined with low household PCI and weak SRs, EALA can occur.

This pathway covers case 11, which is illustrated in Figure 3: Shuai, a 57-year-old villager from Shuangqiao Village, with his wife, has been working as an AL at home for a long time. Shuai suffers from leg disease and cannot be overworked. At present, Shuai manages only 5 mu of arable land, mainly growing rice, rape, and cotton. Now, 1 mu of arable land is conspicuously abandoned. The main reason for this is that most of the local irrigation facilities were built around the 1990s and have been in disrepair for many years, and the irrigation and drainage system are faulty. The arable land contracted by Shuai is in a low-lying area, so it cannot be drained automatically after encountering rain over a long period, which seriously affects the grain output. Moreover, the poor road conditions in the field limit the use of agricultural machinery [58]. It is very inconvenient to transport fertilizers and crops. As a result, EALA occurred.

Pathway 2 shows that even if the ALD is close, the PCI is high, and one of the relatives is a village official, insufficient AL, low-ALT, weak irrigation infrastructure, and poor health conditions can lead to EALA.

This pathway covers case 25, which is illustrated in Figure 3: Li, a 45-year-old villager from Zhongzhou village, has sewing skills. In the 1990s, Li was introduced to work in Guangdong Province by a relative who was a village official. Li′s mother is in poor health and is unable to engage in agricultural activities. Therefore, 6 mu of arable land was planted by a relative in the same village, while the other 2 mu of arable land was abandoned. Despite its proximity to the residential home, the low-ALT and the fact that no one had built irrigation facilities leaves 2 mu of arable land in EALA.

Pathway 3 shows that EALA occurs when ALDs are far, the labor force is insufficient and in poor health, the PCI is low, and social resources are scarce, even if the terrain and ICs of the arable land are good.

This pathway covers cases 1 and 29, which are illustrated in Figure 4: In case 1, Deng, a villager from Huangqiao, is 61 years old. In recent years, due to the death of her husband, the burden of caring for her sick son and arranging agricultural production has fallen on her alone. Two years ago, Deng became unwell, and her ability to work was seriously reduced. Although the village has a flat terrain suitable for mechanized production, most of the production process is still dependent mainly on human labor. Therefore, at present, Deng plants only 3 mu of her arable land, letting her friends and relatives as well as other villagers plant for free on the remaining 9 mu of arable land, but there is still 1 mu of arable land uncultivated. In case 29, Huang, a 78-year-old villager from Dengling, is in poor health after her husband′s death and lives with her son in Xiantao city; none of her family members are engaged in agricultural labor in Dengling. Now, 2 mu of arable land remains abandoned. As in case 1, there is no agreement with friends and relatives in the state of EALA.

The natural environment is the basis for human production activities, arable land use is greatly influenced by natural conditions, and agricultural harvests are inseparable from natural conditions. Therefore, terrain has a significant integrated effect on land use transformation among other natural factors [59]. Moreover, due to the lagging construction of existing agricultural infrastructure and agricultural socialized service systems in China, especially since the abolition of the agricultural tax, arable land water conservancy facilities in the main grain-producing areas have fallen into disrepair over the years, the soil fertility and agricultural labor conditions of arable land have deteriorated, and the ability to resist natural disasters such as drought and flooding has decreased, which greatly affects ALs′ motivation to grow grain. This type of EALA shows strong external characteristics; when the natural and basic conditions of arable land are weak and the ALs themselves have health risks, they choose to abandon arable land that is far away from where they cultivate and has poor natural conditions.

China′s nonagricultural labor remuneration is higher than that of the agricultural sector. According to the survey, the PCI of pure ALs in Louhe town is approximately USD 1548, while that of part-time ALs has reached approximately USD 2322. Moreover, ALs have to bear all kinds of risks in agricultural production and management, but the risks incurred when only participating within the labor force are much smaller. Therefore, middle-aged rural laborers are more willing to choose nonagricultural sectors if they can obtain employment opportunities in those sectors. In addition, health status indirectly affects land use behavior by affecting the quality of labor supply [60]. The damage to ALs′ health reduces their agricultural working time, thus causing a decline in their working ability. This type of EALA shows strong internal characteristics. Even if the cultivated land is flat and has ICs, land far away from home is abandoned when the health of the labor force is poor and the labor force is insufficient in number.

### 4.6. IALA Pathways Analyses

Pathway 4 shows that when the ALD is closer to the residential home, the terrain is flat, ICs are good, agricultural labor is sufficient, PCI is high, SRs are abundant, and ALs have no hidden health problems, IALA ca occur.

This pathway covers cases 2, 8, and 18, which is illustrated in Figure 5: In case 2, Deng is a villager from Dengling village, who is 55 years old, has worked in Guangdong for more than 30 years and has accumulated enough money to start his own business. After much investigation, in September 2018, as a village official, Deng led the registration of vegetable planting ALs’ professional cooperative, with a total investment of more than one million, planting more than 400 mu of vegetables in winter and more than 600 mu of water chestnut in spring and summer. Through the introduction of good varieties of vegetable moss, Deng reduced the cost of raw materials, broadened sales channels, and opened online and offline platforms so that the products could be sold around the country. At the same time, he worked hard in terms of technology and learnt various advanced fertilization and weeding techniques from some experts. Thus, his yield of water chestnuts was much higher than the average level of local ALs. At present, most ALs in the village are members of the cooperative, plant varieties are also expanding, and the total profit is more than USD 123,840. As the IALA of ‘nongrain’ areas is part of planting structure adjustment, it is fundamentally the result of the established constraints of the pursuit of maximum income by the operating entity [61].

In case 8, Chen, a villager from Huangqiao Village, is 56 years old and his granddaughter lives with him and his wife. At present, he operates a total of 60 mu of arable land, of which 54 mu is mainly used for rice cultivation. In recent years, the price of agricultural materials and rural labor have risen rapidly, increasing the cost of grain cultivation, while grain prices have not risen accordingly, and large grain ALs have a strong sense of operational risk and are resistant to increasing their grain production. With the generally low price of agricultural products, Chen planted only one season of rice. Chen noted that the cost input of rice, excluding labor and basic production tools for himself and his wife, involves mainly rent, seeds, fertilizer, pesticides, herbicides, hiring labor, drainage fees and electricity. If other inputs are not considered, then the net income from growing rice is approximately USD 12,384. In addition, because wheat has a slightly lower yield and purchase price, it could either provide a state of slight profit or a greater risk of loss. Therefore, the couple planted rice, according to years of agricultural labor experience, in the winter when wheat should have been planted, as there are more than 40 mu of scattered arable land with no crops planted, exhibiting an IALA phenomenon.

In case 18, Li is a villager of Huangqiao who is 70 years old; his son and daughter-in-law work outside, and he and his wife stay at home to take care of his grandson. Li is the village accountant, which provides him with better social resources, and transferred 10 mu of relatively concentrated paddy fields to plant water chestnut. Compared to the large-scale planting of rice and cotton, sesame, and other crops, planting water chestnut is easier and more profitable. When a batch of water chestnuts is picked and first marketed, a profit of approximately USD 619.2 can be earned after labor accounting for costs. However, because the usual case of the diamond horn is sown in March and April, some early varieties require harvesting at the beginning and middle of July, and by late September to early October, harvesting is over. Therefore, if the return for water chestnut is more satisfactory, in the period of October to March of the following year, the paddy field is usually left idle so that water chestnut can be planted again the next year. However, if the return is less satisfactory, then perhaps the next year, rice or crayfish might be cultivated instead, leading to greater uncertainty. According to Li′s description, in recent years, the planting of water chestnuts has been profitable; therefore, generally, from winter to spring, water chestnut fields are idle, and there is an IALA phenomenon.

Pathway 5 shows that without ALs being plagued by health problems, even if the arable land is close to home, ALs have SRs, if there is a high PCI, the arable land is low lying, and the ICs are poor, IALA still occurs.

This pathway covers case 13, which is illustrated in Figure 5: Wang is a villager from Xinchang village who is 60 years old, relies on his social network resources, has a job delivering goods from his relatives around town, and spends most of his time engaged in agricultural labor with his wife; the rest of the time he is engaged mainly in delivering goods part-time. In the 1980s, Wang had relatives in the village and was a village cadre. Relying on this relationship, Wang contracted a total of more than 20 mu of cultivated land. However, due to the low-lying terrain, the high cost of agricultural inputs and agricultural harvests are not optimistic. Wang has had to devote more time and energy to his part-time job. During the conversation with Wang, it was found that in the process of grain production, the required to water and apply manure has been reduced by more than 30% compared with previous years. The purchase and use of chemical fertilizers and pesticides has increased greatly, which is reflected in Wang′s actual investment in grain management which has decreased, his increased dependence on chemical fertilizers and pesticides, and the abandonment of the traditional intensive cultivation mode of production. During the investigation, Wang stated that ‘it is better to work for two months to grow grain for one year’; the time it takes to weed and fertilize during grain production can be reduced, which is the manifestation of IALA.

It is important to point out that the ‘nongrain’ operation of agricultural scale management subjects, which is supported by the policy in case 2, clearly deviates from the original intention of avoiding an excessive impact on food security. For this reason, it is clearly inappropriate to ‘block’ measures based on food security (i.e., to restrict or prohibit large-scale operators from engaging in nonfood operations), which would not only hinder the development of land-scale operations but would also accelerate the withdrawal of large-scale operators who have already entered the production chain, ultimately leading to greater food security problems. In case 8, large grain growers are shown to have made some contributions to improving land resource utilization and to stabilizing grain production [62]. However, due to being affected by prices and costs, the development of large grain ALs faces a series of problems, such as lagging infrastructure construction, unstable returns from grain cultivation, and low subsidies, and thus, corresponding support measures are urgently needed. In case 18, planting cash crops was illustrated to play an important role in increasing ALs′ income and improving their livelihood, but it also leads to seasonal hidden abandonment, which to a certain extent causes the wasting of arable land resources and threatens the production of food crops. The above type of ALA shows strong internal characteristics, specifically in that ALs rely on their own sufficient labor force, better health, and accumulated social resources and wealth when engaging in agricultural production and operation to overcome the innate deficiencies of natural agricultural conditions. Arable terrain and ICs are no longer the core conditions affecting ALA.

Today’s rural society is still made up of acquaintances and semi-acquaintances, and maintaining interpersonal relationships constitutes a considerable share of the long-term utility function of land use subjects. Social capital, as an informal institution, has an extremely important and embedded role in the allocation of rural land factors. This access to resources indirectly affects the input of agricultural labor households to arable land. Moreover, most grain ALs choose more herbicides and mechanical operations for field management, such as weeding, tillage, fertilization, irrigation, and harvesting, to reduce the required effort, which also leads to the gradual abandonment of traditional agriculture-intensive cultivation techniques and methods developed over many generations in the process of field management, resulting in lower soil fertility, grain yield, and ecological carrying capacity of agricultural land. This type of ALA is the result of a combination of internal and external characteristics, i.e., on the premise that ALs′ intrinsic resources have an advantage so if the natural conditions of arable land are poor then ALs do not invest more time and energy in arable land in pursuit of higher off-agricultural-land income. Therefore, this natural disadvantage perseveres, exhibiting an IALA phenomenon.

### 4.7. Cross-Sectional Analysis

Through an analysis of the five abovementioned pathways, the specific roles of the factors involved in the mechanisms that influence EALA and IALA were also found to be different. Specifically, LH is the core factor influencing EALA, and ICs are an important factor influencing ALs′ EALA. When comparing pathways 1, 2, and 3, the common points are that the natural agricultural conditions of the land are poor, such as the low-lying terrain and poor irrigation facilities, and that the ALs face health hazards and cannot engage in too many agricultural activities. The difference is that the former factor is constrained by scarce socioeconomic resources and the disadvantage of poor external environmental conditions cannot be changed, while the latter presents a better socioeconomic situation but insufficient agricultural labor and the external environment cannot be changed under the existing circumstances.

The intrinsic conditions of health, agricultural labor, PCI and SRs provide the core factors affecting the hidden abandonment of agricultural labor households. When comparing pathway 3 and pathway 4, the common point to be found is that the agricultural labor force is more abundant, social resources are abundant, the economic base is better, and there are no obvious health problems. The difference is that the former uses existing social connections and capital to continuously invest in agricultural production and operation, while the latter uses social connections to engage in nonagricultural work. Although the results are both IALA, they are substantially different. The former has increased its input to agriculture, while the latter has reduced its input to agriculture.

PCI and SRs are two-sided for ALA. When comparing pathways 1 and 3, it can be seen in pathway 3 that ALs′ SRs and economic conditions are relatively more abundant, and they are not affected by health problems. In this case, the natural conditions of arable land are good or bad, and even if the natural conditions are poor, ALs tend to continue agricultural production due to the large scale of agricultural production and operation in which they are engaged and which provide certain benefits. In general, the natural conditions of arable land have little effect on ALA behaviors. However, to a certain extent, this is also due to the pursuit of higher returns; ALs tends to cultivate nongrain, resulting in IALA.

In both pathway 1 and 4, ALs are constrained by the external conditions of the cultivated land, in that it is low lying and has weak irrigation facilities. The difference is that in pathway 1, ALs have a poor socioeconomic base and their own health conditions, but in pathway 4, they have better SRs and higher PCI. Both pathways 1 and 4 led to ALA, but ALs in pathway 1 presented more passivity than those in pathway 4. To a certain extent, ALs in pathway 1, without capital accumulation and social connections, may engage in agriculture as their only option and may replant existing abandoned arable land if the arable land infrastructure is more complete and no longer constrained by terrain and ICs. ALs in pathway 4, conversely, exhibit a higher level of initiative, which is reflected in their tendency to obtain more off-agricultural-land benefits by means of social connections.

By comparing pathways 2 and pathways 4 and 5, it can be seen that in pathways 2 and 5, agricultural households rely on better social resources to engage in nonagricultural work and to earn income. However, because the ALs in pathway 2, in the existing situation, work from outside the home, the labor force that remains at home is insufficient and in poor health, leading to EALA, while those in pathway 5 are not affected by a labor shortage and health problems but by reduced agricultural inputs, leading to IALA. It is worth noting that the ALs in pathway 2 who work outside the home can make up for the shortage in labor and the deterioration in its quality by choosing to return home and work in agriculture. However, the situation may be the same as that of the ALs in pathway 4, who used social connections and capital accumulated in the early stage to engage in large-scale production and operation and to plant cash crops.

## 5. Conclusions

Although this study is exploratory given the limited sample size, it reveals the characteristics of the studied EALA and IALA, some cases of which are related to ALD, ALT, ICs, ALs, LH, PCI, and SRs. Three possible pathways that contribute to EALA and two pathways that lead to IALA are presented.

An analysis of the configurations sufficient for EALA shows that health is a decisive condition; it appears in all three configurations that lead to EALA, which means that ALs with poorer health status choose EALA, revealing that good physical health is an important prerequisite for engaging in agricultural production, and diseases significantly reduce the ability of rural middle-aged and elderly people to participate in agricultural work. Moreover, the impact of the natural environment and agricultural infrastructure should not be underestimated. For most ALs, the cost of remediating arable land is high, and therefore, when arable land is flooded due to low-lying conditions, they choose not to plant any crops to reduce costs.

An analysis of the configurations sufficient for IALA shows that LH, ALs, PCI, and SRs are the core factors as they appear in both configurations that lead to EALA, which means that when the following conditions exist: ALs are healthy, there is sufficient household ALs, a high PCI and abundant SRs, ALs choose IAIA. This finding can explain the current reduction in arable land inputs and the dominant choice of non-grain crops by most ALs in major food-producing regions. Moreover, ALs seek nonagricultural opportunities using their abundant social resources to obtain higher off-agricultural-land returns. In addition, ALs can develop special industries through better endowments to obtain higher agricultural returns compared to returns from growing grain. Both forms lead to a decrease in inputs to the grain industry and affect food security.

Additionally, this study offers two main recommendations for national policy makers. First, given that the pathways that lead to EALA and IALA are different, the measures taken by policy makers to promote EALA should differ from those taken to promote IALA. For example, this study shows that health is a core condition and that terrain and irrigation are the main conditions contributing to EALA. Thus, if policy makers aim to alleviate EALA, paying attention to rural health issues and improving production conditions may be appropriate measures. However, if policy makers aim to address IALA, then they should improve agricultural returns and strengthen food security publicity and guidance, thereby promoting the rational use of arable land. Second, the results show that ALA activities are influenced by the interdependence of multiple institutional conditions, and that equally effective pathways exist. Therefore, policy makers should choose a pathway suitable for their country’s actual conditions and allocate their limited resources to improving the institutions that are currently critically in need of assistance based on the pathway they have selected.

There are limitations to this study. For example, our findings emerge from the specific context of the main grain-producing areas. Due to the difference in the natural environment in the main grain-producing areas, the planting structure and multiple cropping index have regional differences. In identifying the influencing mechanisms of ALA, the influence degree of each factor should be slightly different. Therefore, in the future, our work will explore the mechanism from different regions and perspectives in combination with the natural geographical environment, agricultural production characteristics, social and economic conditions, and other factors in different regions. However, this study reflects processes that are common across many locales. Clearly, the identified pathways apply most directly to core grain-producing areas in China. Nonetheless, our broader findings about how ALD, ALT, ICs, ALs, LH, PCI, and SRs affect ALA are relevant elsewhere in China and in other settings. The way these elements converge elsewhere depends on the varying natural resources, cropping systems, household characteristics, and agricultural infrastructure. Beyond China’s borders, we can expect all the social, economic, and environmental factors that have been shown to influence EALA and IALA to also have an effect. In line with recent moves towards a diagnostic approach to the study of environmental governance, the in-depth comparative approach adopted in this study can be a useful tool for specifying how ALs carry out EALA and IALA.

## Figures and Tables

**Figure 1 ijerph-19-05090-f001:**
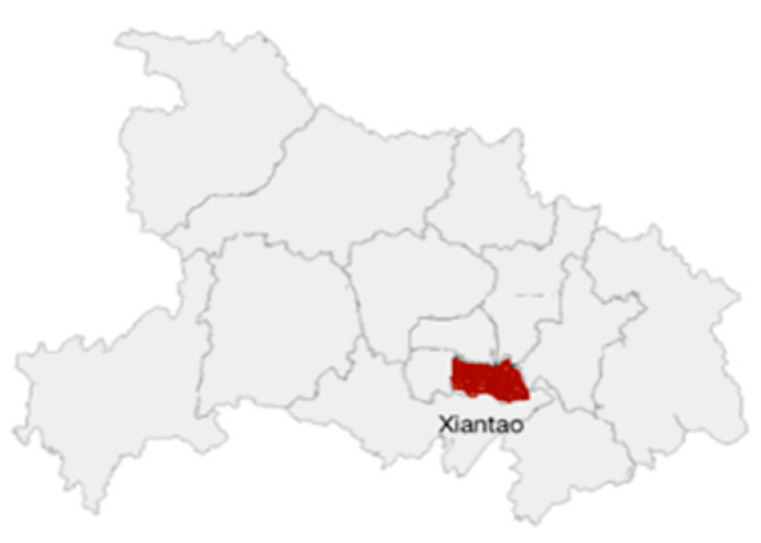
Location of Xiantao city in Hubei province.

**Figure 2 ijerph-19-05090-f002:**
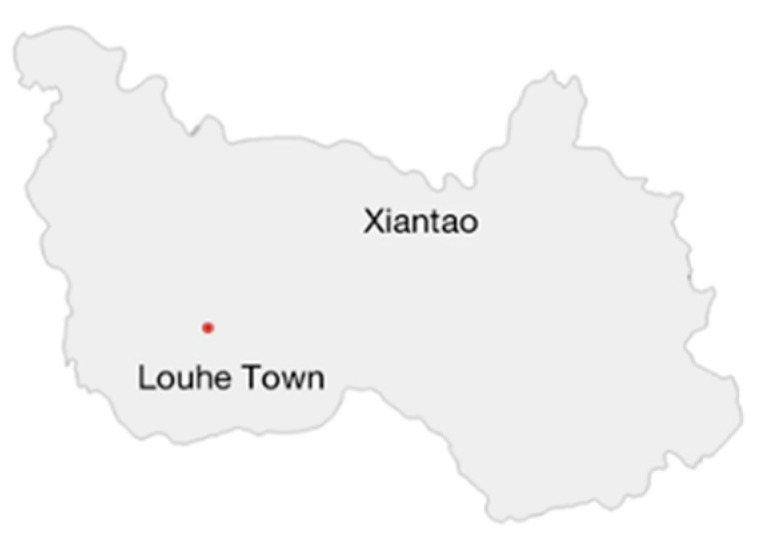
Location of the study area in Xiantao city.

**Figure 3 ijerph-19-05090-f003:**
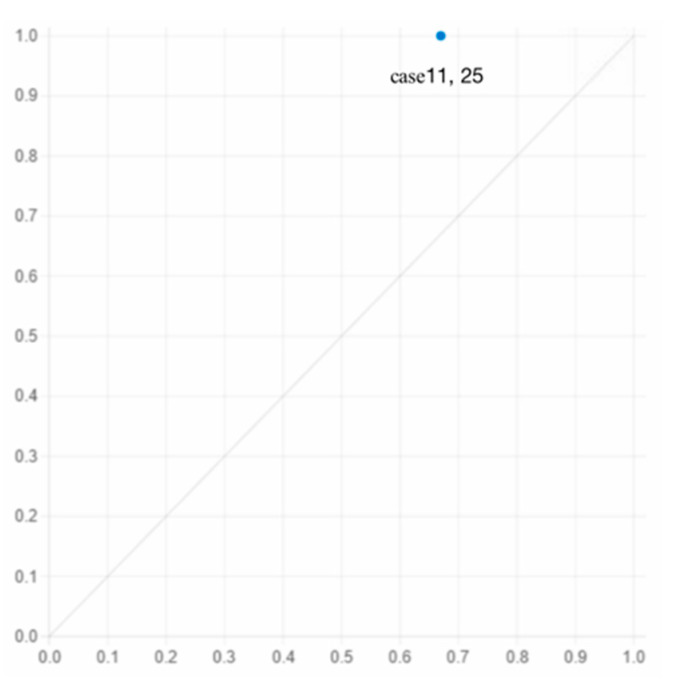
Explanation case of pathway 1 and pathway 2.

**Figure 4 ijerph-19-05090-f004:**
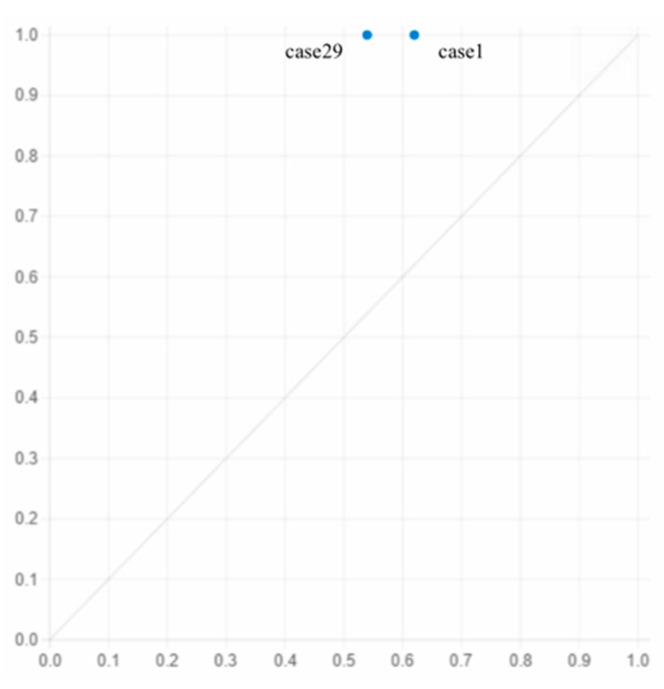
Explanation case of pathway 3.

**Figure 5 ijerph-19-05090-f005:**
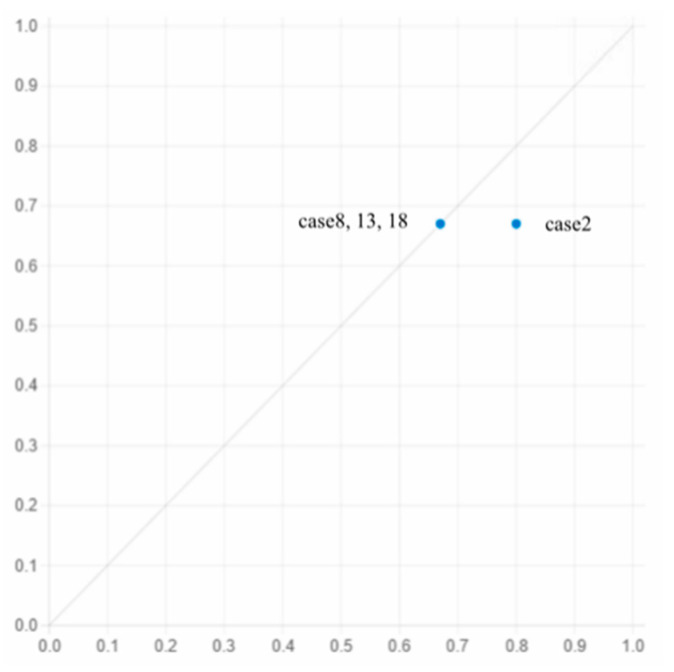
Explanation case of pathway 4 and pathway 5.

**Table 1 ijerph-19-05090-t001:** Fuzzy membership calibrations.

Variable		Fuzzy Membership Score		
		Full Nonmembership	Crossover Point	Full Membership
Conditional variables	ALT	0, 0.33, 0.67, 1		
	ICs	0, 0.33, 0.67, 1		
	LH	0, 0.33, 0.67, 1		
	SRs	0, 1		
	ALD	59.45	367.5	914
	ALs	0	1.2	2
	PCI	878.5	2732.2	5776
Outcome variable	ALA	0, 0.33, 0.67, 1		

Notes: ALT stands for arable land terrain. ICs stands for irrigation conditions. LH is laborer health, and SRs is social relationships. ALD is arable land distance. ALs is an agricultural laborer. PCI is household per capita income. According to the exchange rate during our investigation, we changed the statistical unit of PCI from RMB to USD. ALA is arable land abandonment.

**Table 2 ijerph-19-05090-t002:** Truth table of factors influencing ALA.

Case	ALD	ALT	ICs	ALs	LH	PCI	SRs	ALA
Case 1	0.73	0.33	0.33	0.38	1	0.01	0	1
Case 2	0.2	0	0	1	0	1	1	0.67
Case 3	0.52	0.33	0.67	0.95	0	0.04	1	0
Case 4	0.8	0.33	0.33	0.95	0	0.24	0	0.67
Case 5	0.06	0.33	0	0.95	0.33	0.14	0	0.33
Case 6	0.75	0	0.33	0.15	0	0.52	1	0.67
Case 7	0.88	0.33	0.33	0.75	0	0.1	0	0
Case 8	0.18	0	0.33	0.95	0.33	0.92	1	0.67
Case 9	0.09	0.33	0.33	0.95	1	0.55	1	0
Case 10	0.57	0	0.67	0.95	0.33	0.43	0	0.67
Case 11	0.94	1	1	0.95	0.67	0.18	0	1
Case 12	0.11	0.67	0	0.95	0	0.07	1	0
Case 13	0.09	0.67	1	0.95	0	0.9	1	0.67
Case 14	0.47	0.33	0	0.95	0.33	0.57	0	0
Case 15	0.77	0	0.33	0.15	0	0.51	0	0.67
Case 16	0.05	0.67	0.33	0.95	0.33	0.46	0	0.33
Case 17	0.05	0	0.33	0.38	0.33	0.33	0	0
Case 18	0.18	0.33	0.33	0.95	0.33	0.86	1	0.67
Case 19	0.15	0	0.33	0.38	0	0.59	0	0.67
Case 20	0.07	0.67	0.67	0.5	0	0.15	0	0
Case 21	0.97	0.33	0.67	0.38	0	0.16	0	1
Case 22	0.96	0	1	0.5	0.33	0.06	0	1
Case 23	0.45	0	0.67	0.38	0	0.49	0	0
Case 24	0.05	0.33	0	0.38	0	0.54	0	0.67
Case 25	0.05	1	1	0.05	0.67	0.96	1	1
Case 26	0.92	0	0.33	0.05	0	0.84	0	1
Case 27	0.09	1	1	0.08	0	0.81	1	0
Case 28	0.85	0.33	0.33	0.05	0.33	0.72	0	0.33
Case 29	0.78	0.33	0.33	0.05	0.67	0.46	0	1
Case 30	0.05	0.67	0.33	0.05	0	0.94	1	0.33

**Table 3 ijerph-19-05090-t003:** Necessity of conditions.

Condition	Consistency	Coverage
ALD	0.617	0.723
~ALD	0.527	0.461
ALT	0.397	0.579
~ALT	0.778	0.594
ICs	0.597	0.674
~ICs	0.623	0.560
ALs	0.546	0.481
~ALs	0.548	0.636
LH	0.354	0.762
~LH	0.778	0.507
PCI	0.620	0.640
~PCI	0.581	0.565
SRs	0.312	0.425
~SRs	0.688	0.544

Notes: ‘~’represents the absence of conditions.

**Table 4 ijerph-19-05090-t004:** EALA and IALA pathways.

Conditional Variable	EALA Pathways	IALA Pathways
H1	H2	H3	H4	H5
ALD	●	○	●	○	○
ALT	●	●	○	○	•
ICs	●	●	○	○	•
ALs	•	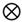	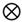	●	●
LH	●	●	●	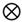	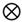
PCI	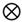	•	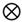	●	●
SRs	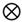	•	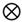	●	●
Consistency	1	0.935	0.941	0.902	0.902
Raw coverage	0.077	0.048	0.106	0.147	0.073
Unique coverage	0.045	0.041	0.074	0.122	0.045
Overall solutionconsistency			0.925		
Overall solutioncoverage			0.387		

Notes: ‘●’ represents the presence of the core causal condition, ‘
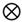
’ represents the absence of the core condition, ‘•’ represents the presence of the auxiliary condition, ‘○’ represents the absence of the auxiliary condition, and ‘blank’ means that the condition can either appear or not appear in the configuration.

## Data Availability

Data are provided within the article.

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
