# Peer review of "What Drives Land Abandonment in Core Grain-Producing Areas? Evidence from China"

_ijerph, 2022, doi:10.3390/ijerph19095090_

Round 1

Reviewer 1 Report

The article is very interesting both in terms of content and observation. Extensive and up-to-date literature on the subject indicates a very good preparation of the authors for the analysed subject. I just put forward some minor suggestions.

  1. The introduction part (Line no. 55 to 57) mentioned that "Although some scholars have studied the factors that lead to ALA, few studies have examined the effects of different combinations of factors on the EALA and IALA of agricultural households." but no references have been used. It is recommended that similar studies are mentioned and compared to own research results. 
  2. In terms of theoretical analysis (Line 92), the following definition of abandoned arable land may be more appropriate: “Agricultural land abandonment is often referred to as the cessation of farming and giving away land for natural successions, such as grasses, shrubs, and trees on former agricultural lands, but may also result in land degradation.” Source: https://www.oxfordbibliographies.com/view/document/obo-9780199363445/obo-9780199363445-0129.xml
  3. The sample size is quite small when compared to the population size. The study only collected 30 samples, which may not represent the entire population. 
  4. For an international audience's understanding, USD should be used instead of Yuan (CNY) and Renminbi (RMB) (e.g. 15,000 yuan, 10,000 yuan, 800,000 yuan, 80,000 RMB) in the manuscript. 
  5. As part of good research practice, the author should mention the limitations of their research and what further research is needed.

Author Response

Thank you for your comments and suggestions. We have supplemented relevant literature. For more details, please see reference list [12] - [16] and Line 55-57 in the text. According to the book "Agricultural Land Abandonment" you recommended, we found a key important reference (in the reference list [8]) in this book and redefined the concept Line 96-97. See the reference list for details [8]. We have changed the RMB to USD according to the exchange rate in our survey and revised the corresponding figures. We also added the shortcomings of this study and the further work in the conclusion part. All of these revisions can be traced by switching between 'Simple Mark-up' and 'All Markup'  in the column of Review button in the Word.

Reviewer 2 Report

Comments to Authors:

The article is generally written well. The data available in the paper has sufficient value and practical applications. However, some important points need attention before the publication. Below I have provided few comments which I hope will improve the manuscript.

Introduction

Line 29: Please improve the sentence.  May be consider to replace the sentence

“Arable land abandonment (ALA) is a global change in land use.”  by “Arable land abandonment (ALA) is one of the dominant processes of change in rural areas all over the world.”

Line 49: abbreviations should be defined at first use.

Methods

Lines: 246-249: It is not clear how the case study households were selected. The sampling design employed to select the surveyed households should be clearly defined and justified.

Line 375: add “s” where suggested.

Line: 415: missing from the References list. Please check.

Lines: 448, 465 and 565: the figures are not visible.

Author Response

Thank you for your comments and suggestions. We have supplemented relevant literature. All of these revisions can be traced by switching between 'Simple Mark-up' and 'All Markup'  in the column of Review button in the Word. For the figures in the text Lines 448,465 and 565, we are trying our best to make them more clearly. 

Reviewer 3 Report

The document brings up a topic of great interest, namely the abandonment of arable land. 
This phenomenon has many repercussions, and its analysis and the attempt to establish different methods of reducing it and improving land use are more than welcome.
The study carried out in China, a major player in the global agricultural market, can help researchers in other areas facing the same phenomenon. 

Author Response

Thank you for your comments and suggestions. We have supplemented relevant literature. All of these revisions can be traced by switching between 'Simple Mark-up' and 'All Markup'  in the column of Review button in the Word. 
